# Adaptation and Validation of the Pet Bereavement Questionnaire (PBQ) for Chinese Population

**DOI:** 10.3390/ani14192845

**Published:** 2024-10-02

**Authors:** Winnie W. Y. Yiu, H. N. Cheung, Paul W. C. Wong

**Affiliations:** Department of Social Work and Social Administration, The University of Hong Kong (HKU), The Jockey Club Tower, The Centennial Campus, Pokfulam, Hong Kong; wyyiu929@hku.hk (W.W.Y.Y.); amy.hn.cheung@hku.hk (H.N.C.)

**Keywords:** animal bereavement, grief, depression, companion animal, loss, psychometric validation

## Abstract

**Simple Summary:**

Pet ownership is growing in Chinese societies, but tools to assess grief from pet loss among the Chinese are lacking. This study adapted and validated the Pet Bereavement Questionnaire (PBQ) in Hong Kong, which aimed to provide a culturally appropriate assessment tool for pet bereavement. The findings supported a three-factor structure—grief, anger, and guilt—similar to the original PBQ. The validated Chinese version of the PBQ demonstrated strong reliability, meaning it consistently measures what it is supposed to, and strong validity, meaning it accurately reflects the emotional experiences of pet loss. This tool can significantly enhance our understanding of the emotional complexities surrounding pet loss within this cultural context, offering critical insights that can guide more effective support interventions and public health strategies for individuals experiencing the often-unrecognized grief associated with pet loss.

**Abstract:**

Despite the increasing prevalence of pet ownership in Chinese societies, standardized tools to assess grief from pet loss remain lacking. Research predominantly focuses on Western populations, creating a gap in understanding pet bereavement in Chinese cultural settings. This study aimed to adapt and validate the Pet Bereavement Questionnaire (PBQ-C) for a Chinese context to create a culturally appropriate assessment tool. A total of 246 participants with companion animal loss experiences were recruited through the university of the research team. They were invited to complete an online survey including the PBQ-C, the Depression subscale of the Depression Anxiety Stress Scales (DASS-21), and the Inventory of Complicated Grief (ICG). Both Exploratory Factor Analysis and Confirmatory Factor Analysis were conducted to examine the psychometric properties of the PBQ-C and the findings supported a three-factor structure—grief, anger, and guilt—aligned with the original PBQ, with three items reassigned to different factors. Despite these adjustments, the PBQ-C demonstrated strong internal consistency, reflecting the reliability of the questionnaire in measuring the same construct across its items; split-half reliability, indicating its ability to produce consistent results when divided into two parts; and concurrent validity, showing that the PBQ-C correlates well with other established measures of grief. The validated PBQ-C provides a culturally sensitive tool for assessing pet bereavement in Chinese society that can promote research and counselling support for this under-researched and under-recognized type of loss of human-animal relationships.

## 1. Introduction

Owning pets has become increasingly popular across Chinese societies in Mainland China, Taiwan, and Singapore [1,2,3]. In Hong Kong, pet ownership has risen 72 percent from 297,100 dogs and cats in 2005/6 to 510,600 in 2015/16 and 545,600 in 2019 [4], but these figures are likely to be underestimated as they excluded companion animals in non-residential spaces and those unreported by households. The patterns of companion animal ownership are likely to have changed recently given broad social shifts and the impact of COVID-19. During the pandemic, pets played an important role in offering emotional and psychological support, such as providing companionship to elderly residents [5,6]. There is also a growing trend in pet ownership among young couples and families and recent anecdotal records have shown that many young married couples are delaying or opting out of parenthood and see their companion animals as their children [7,8].

Research that specifically studies the well-being of pet owners found that companion animals positively facilitate adult attachment, emotion regulation, and mental well-being, supplementing attachment needs that might be lacking in human relationships [9]. Regarding social well-being, companion animals may act as social facilitators in enhancing social connectedness within communities [10]. Companion animals have been regarded by some as a source of unconditional love and something that can fulfill human needs for care, companionship, and physical contact [11]; for these reasons, companion animals are easily regarded as integral members of the family unit.

When companion animals are regarded as family members, the experience of losing a companion animal can be very distressing for those families. In fact, research has found that the intensity of grief following the loss of a pet can be significant, with many individuals experiencing symptoms akin to those of losing a close human relative [12]. Grief from pet loss can last from six months to a year, or even longer [13], and the duration and intensity of grief are closely related to the level of attachment of the pet owners [14,15]. For those who integrate pets deeply into their life, the loss of a pet can lead to complicated grief [16] and symptoms of post-traumatic stress disorder (PTSD) including persistent longing, intrusive thoughts, and difficulty moving forward [16]. A recent study examined individuals who had been grieving for over a year and experiencing functional impairment due to pet loss and found that their grieving processes aligned with the DSM-5 model of grief, indicating that similar symptoms of prolonged grief disorder, as described in the DSM-5, can apply to pet bereavement. [17].

This grief of losing a pet is commonly seen as a form of disenfranchised grief that is minimized or unrecognized by society. Research indicates that many individuals grieving the loss of their pets face a lack of social support and recognition, which can intensify negative emotions and prolong the mourning process e.g., [18,19]. In the Chinese context, there is a cultural tendency to suppress grief, largely influenced by the emphasis on stoicism and emotional restraint, which may further exacerbate the experience of internalized grief in pet loss [20,21]. However, no empirical literature seems to have supported this hypothesis, as research in exploring pet bereavement within Chinese societies was very limited. To the best of the authors’ knowledge, there has only been one qualitative study conducted in Hong Kong that explored grief reaction and post-traumatic growth—the positive psychological changes experienced as a result of struggling with highly challenging life circumstances—in the context of companion animal loss [21]. The absence of validated psychometric tools in Chinese contexts has hindered empirical research. The unique cultural and social dynamics influencing the grieving processes in these regions remain underexplored.

To address this gap, our study aimed to adapt the Pet Bereavement Questionnaire (PBQ) into a culturally appropriate tool for the Chinese context. The PBQ is a brief and well-validated tool designed to assess psychological responses following the loss of a pet [22]. Although there is another scale examining reactions after pet loss [23], the PBQ is chosen due to its versatility in assessing grief across various types of pets and was widely validated and successfully adapted in different cultural contexts [24,25].

We hypothesized that the Chinese PBQ (PBQ-C) would (H1) demonstrate strong internal consistency, meaning the items would reliably measure the same underlying construct; (H2) have strong concurrent validity, indicating that the PBQ-C would correlate well with other established measures of grief; and (H3) have a three-factor structure in line with the original PBQ developed and mainly validated in non-Asian societies (H3). With this study, we hoped to provide a culturally sensitive tool for assessing pet bereavement in Chinese society, enabling more empirical research, and facilitating evidence-based clinical interventions associated with pet loss within this population.

## 2. Materials and Methods

### 2.1. Participants

Participants were selected using convenience sampling. The online, self-administered survey was circulated through posters around the campus, and through internal emails that could reach over 35,000 students and staff at the university of the research team and sharing through social media platforms. The inclusion criteria required participants to be aged 18 or above, provide informed consent, and have experienced pet loss. No incentives were offered for participation. Between 11 December 2023 and 19 January 2024, a total of 301 participants were reached, with 264 completing the study. After data cleaning, 18 participants who had not experienced pet loss were excluded, resulting in a final sample of 246 participants.

### 2.2. Materials

Data were collected using measures specifically adjusted to reference the deceased pet. Participants were instructed to focus on their most recent pet loss and to consistently reference this pet when answering all survey questions.

#### 2.2.1. Demographics Information

This questionnaire included questions about the participant’s age, level of education, occupation, marital status, monthly personal income, number of child(ren), number of family member(s), whether they currently have pets, number of pets, species of pet, time passed since the death, cause of death, and whether the deceased pet had undergone euthanasia.

#### 2.2.2. Pet Bereavement Questionnaire (PBQ)—Chinese Version

The Pet Bereavement Questionnaire (PBQ) was originally developed by Hunt and Padilla [22] to assess grief reactions related to the loss of a pet. The PBQ is a 16-item scale structured around the following three factors: grief (7 items), anger (5 items), and guilt (4 items). Items are rated on a 4-point Likert scale (0 = Strongly disagree, 3 = Strongly agree), with higher scores indicating greater levels of grief symptoms. The original PBQ demonstrated good internal reliability (Cronbach’s α = 0.87) and strong construct validity. In this study, the PBQ-C was developed by translating and adapting the original scale. The translated items remained identical to those in the English version and retained the original three-factor structure, except for the inclusion of an additional “not applicable” option for the first item. The modification aimed to ensure the scale’s relevance across diverse pet ownership experiences in Hong Kong. As many exotic pet owners in Hong Kong face challenges in accessing specialized veterinary care due to a scarcity of facilities and high treatment costs [26], direct interactions with veterinarians in critical situations are uncommon, making the original item less applicable.

#### 2.2.3. Depression Subscale of the Depression, Anxiety, and Stress Scale (DASS-21)—Chinese Version

The Depression subscale of the DASS-21 was originally developed by Lovibond and Lovibond [27] to assess depression, anxiety, and stress in adults. This subscale consists of seven items that assess the level of depression by asking participants to rate the frequency and severity of symptoms experienced over the past week on a 4-point Likert scale (0 = did not apply to me at all, 3 = applied to me very much or most of the time). Higher scores indicate higher levels of depressive symptoms. The subscale demonstrated excellent internal consistency, with a Cronbach’s alpha of 0.92, indicating high reliability. In the Chinese version, the DASS-21 Depression subscale was adapted and validated with a strong internal consistency of Cronbach’s α = 0.91 and good test–retest reliability [28].

#### 2.2.4. Inventory of Complicated Grief (ICG)—Chinese Version

The Inventory of Complicated Grief (ICG) was originally developed by Prigerson, et al. [29] to assess symptoms of Prolonged Grief Disorder (PGD) in adults. The ICG consists of 19 items measured on a 5-point Likert scale (0 = Never, 4 = Always) to evaluate the severity and persistence of grief-related symptoms. Scores of 0–2 are considered “endurable”, while scores of 3–4 are deemed “intolerable”, indicating a higher risk of complicated grief. A higher score represents a higher intensity and complexity of grief. The scale demonstrated high internal consistency with a Cronbach’s alpha of 0.93 and high test-retest reliability. The Chinese version of the ICG was adapted and validated for use in the Chinese population [30]. The scale demonstrated excellent internal consistency, with a Cronbach’s alpha of 0.94, indicating high reliability in the present study.

### 2.3. Translation and Adaptation Procedure

Permission was requested from the authors of the original PBQ for translation in Traditional Chinese. We adhered to the translation process recommended by the guidelines for cross-cultural research [31]. First, two bilingual researchers independently translated the English version of the PBQ into Traditional Chinese. Discrepancies between the two translations were discussed and resolved collaboratively, leading to a unified Chinese version. The translated items were reviewed and compared with the original English version by several researchers and a psychologist with a Doctorate in Clinical Psychology, specializing in human-animal interactions and mental health research, to ensure both linguistic accuracy and cultural relevance. Back-translation was carried out by a different set of bilingual researchers who were blind to the original scale. The back-translated version was then compared with the original English PBQ to identify and resolve any inconsistencies. The final version of the PBQ-C was determined after consensus was reached by the research team.

### 2.4. Data Analysis

The data analysis of the reliability and validity of the PBQ-C was conducted using statistical methods and software packages mentioned below, primarily utilizing SPSS 25.0 and AMOS 23.0 at a 5% significance level. The “not applicable” option in item 1 was coded as a “missing value” in the scale. The total scores for these participants were calculated based on the remaining 15 items, the mean scores of the Anger sub-scale were calculated without the first item. Pairwise analysis was chosen to minimize data loss during the analysis.

Multiple analyses were conducted to assess the psychometric properties of the PBQ-C. In terms of validity, an Exploratory Factor Analysis (EFA) using Principal Axis Factoring (PAF) was conducted to explore the underlying factor structure. Sampling adequacy was confirmed by the Kaiser-Meyer-Olkin (KMO) measure to determine the appropriateness of factor analysis, with values suggesting adequacy. Factors were extracted based on the Kaiser criterion, which suggested retaining factors with eigenvalues greater than 1. An oblique rotation (Direct Oblimin) was utilized to facilitate interpretation of the factor structure, allowing correlations between factors. The threshold for significant factor loadings was set at 0.30 to ensure satisfactory factor representation [32].

To further validate the factor structure identified by the EFA, a Confirmatory Factor Analysis (CFA) was performed using SPSS AMOS 23.0 to assess the validity of the item distribution. It is a theory-driven technique which determines the goodness-of-fit between the model and the sample data [33]. There are a few fit indices used in this study to discern how well the specified model reproduces the covariance matrix among the indicator items [34]. They are grouped under following four main groups of measures: practical fit measures (chi-square statistics or χ^2^/df), absolute fit indices (GFI, RMSEA), incremental fit indices (NFI, CFI), and Akaike’s Information Criterion (AIC) measures to predict the accuracy of the model. A good model fit is indicated by a χ^2^/df ratio less than 3, CFI and NFI values close to or greater than 0.95, a GFI value greater than 0.90, an RMSEA less than 0.06, and lower AIC values relative to the competing models [32]. The degrees of freedom (χ^2^/df) were considered instead of the chi-square statistic due to small sample size [35].

The concurrent validity of the PBQ-C was assessed by examining the Pearsons correlations between each PBQ subscale, the total score of PBQ, the Inventory of Complicated Grief (ICG) scale, and the Depression Anxiety Stress Scale’s (DASS-21) depression subscale.

In terms of reliability, internal consistency reliability was assessed using Cronbach’s alpha and Guttman’s split–half reliability to confirm the robustness across the entire scale and its subscales. The split-half reliability was calculated by dividing the scale into odd and even-numbered items.

Spearman’s rank-order correlations and non-parametric tests, including the Mann-Whitney U test and the Kruskal-Wallis H test, were employed to explore the relationships between PBQ scores and demographic variables such as age, number of pets, type of pet, and cause of death. Non-parametric analyses were chosen because the PBQ score distributions did not meet the assumptions of normality, as indicated by skewness in the data although the sample size was not small. These analyses aimed to identify any significant associations between demographic characteristics and the experience of pet bereavement. To assess differences in grief intensity between different types of pets, dogs and cats were grouped into a single category because they were the two most common types of companion animals in Hong Kong and compared with all other pet types. This grouping was based on prior research indicating that dogs and cats typically form more significant emotional bonds with their owners compared to other pets [36,37], potentially leading to higher grief intensity.

## 3. Results

### 3.1. Demographics

The sample included a total of 246 participants with a mean age of 34.93 years (SD = 12.00). Regarding personal monthly income, the average income was HKD 25,370 (SD = 20,564), with 18.3% reporting no income and 22% earning between HKD 20,000 and HKD 29,999. In terms of family structure, participants had an average of 1.82 family members living with them (SD = 1.17), and the majority (67.1%) had no children, with the average number of children being 0.51 (SD = 0.62). Participants reported having an average of 2.73 pets throughout their lives (SD = 1.69). The time since their pet passed away ranged from 1 to 24+ months, with an average of 19.45 months (SD = 8.19).

Missing values were minimal and primarily due to participants skipping certain questions. These missing responses are assumed to be random and are not expected to bias the results. Detailed demographic characteristics of participants and their deceased pets are presented in Table 1.

### 3.2. Validity

#### 3.2.1. Construct Validity

The EFA confirmed that the PBQ-C retained a similar factor structure to the original, with three primary factors—grief, anger, and guilt—accounting for 55% of the total variance (Table 2). The Kaiser-Meyer-Olkin (KMO) measure was 0.92, and Bartlett’s Test of Sphericity was significant (χ^2^ = 1836.29, *p* < 0.001), indicating strong data suitability for factor analysis. While most item loadings corresponded to the original structure, three items were reassigned based on cultural and statistical considerations.

First, item 4, “I have had nightmares about my pet’s death”, loaded onto the guilt factor (0.43) rather than the original anger factor (0.03). Second, item 14, “Memories of my pet’s last moments haunted me”, loaded onto the grief factor (0.54) instead of the anger factor (0.07). Lastly, item 15, “I’ll never get over the loss of my pet”, loaded onto the anger factor (0.70) rather than the grief factor (0.36).

These items were reassigned in the PBQ-C based on both content analysis and the statistical outcomes. Table 2 presents all items with corresponding factor loadings. The factor communalities for all items in the scale were found to be 0.33 or higher, suggesting that the factor loadings were adequate for the Chinese PBQ.

The following two models were compared by CFA: Model 1 (original factor distribution) and Model 2 (revised factor distribution). Model 2 demonstrated better fit across all indices, including lower RMSEA and higher CFI, confirming that the revised model better represents the data (Table 3). The standardized factor loadings for all items exceeded 0.30, and the correlations between the latent factors were within expected ranges, highlighting the interrelated yet distinct emotional dimensions of grief, anger, and guilt (Figure 1).

#### 3.2.2. Concurrent Validity

Significant positive relationships were found between the PBQ and both the ICG and DASS-21 depression subscales (Table 4). These correlations ranged from moderate to strong, supporting the PBQ’s ability to measure grief and depression in pet loss contexts.

### 3.3. Reliability

#### 3.3.1. Internal Consistency Reliability

The PBQ-C demonstrated strong internal consistency, with Cronbach’s alpha values of 0.92 for the total scale, 0.90 for grief, 0.79 for guilt, and 0.76 for anger. Item-total correlations, corrected for overlap, ranged from 0.46 to 0.75, with an average of 0.62, suggesting that all items are conceptually cohesive while capturing distinct aspects of pet bereavement (Table 5).

#### 3.3.2. Split-Half Reliability

The odd numbered items were calculated as 0.82 and second half—the even numbered items were calculated as 0.88. The Guttman split-half reliability coefficient was 0.86, indicating that the items maintained high reliability under split-half conditions.

### 3.4. The Relationship between PBQ and Demographic Variables

Spearman’s Correlation revealed a weak negative correlation between age and guilt subscale (r_s_ = −0.10, *p* < 0.001), but no significant correlation between other sub-scales and total PBQ. Age was not significantly correlated with DASS-21’s depression subscale, the level of depression did not account for the findings. The number of pets was weakly correlated with anger (r_s_ = 0.15, *p* < 0.05) and grief (r_s_ = 0.13, *p* < 0.05).

The Mann–Whitney U test showed higher total PBQ scores among dog and cat owners (group 1; median = 29) compared to owners of other pets (group 2; median = 27) (U = 4210.50, z = −2.13, *p* = 0.03), indicating more intense grief for the dog and cat owner group. Similar results were found in the grief subscale, with dog and cat owners reporting slightly higher scores (group 1; median = 14; group 2; median = 12) (U = 4210.50, z = −2.13, *p* = 0.03).

There was a significant difference in depression scores between individuals who had experienced euthanasia of their pets (median = 1) and those who had not (median = 4) (U = 7907.50, z = 2.18, *p* = 0.03), with individuals who had experienced euthanasia of their pets reporting lower depression levels.

The Kruskal-Wallis H test revealed significant differences in PBQ total scores (χ^2^/(4) = 32.24, *p* < 0.001), as well as in the guilt (χ^2^/(4) = 18.82, *p* < 0.001), anger (χ^2^/(4) = 37.80, *p* < 0.001), grief (χ^2^/(4) = 19.64, *p* < 0.001), ICG (χ^2^/(4) = 33.31, *p* < 0.001), and depression scores (χ^2^/(4) = 20.03, *p* < 0.001) across different causes of death. Generally, poisoning/attack and accidents were associated with higher scores across most subscales, while health issues related to old age tended to result in lower scores. Notably, the anger subscale showed the highest median score for cases of medical negligence or complications after treatment (median = 2), whereas the lowest scores were observed in deaths due to old age (median = 1).

## 4. Discussion

### 4.1. Interpretation of Results

This study is the first attempt to back-translate and examine the translated Traditional Chinese version of the PBQ in Hong Kong. In sum, the findings provide evidence supporting the strong psychometric characteristics of the PBQ-C for use with a Chinese population in Hong Kong. The internal consistency and reliability of the PBQ-C were confirmed through high Cronbach’s alpha coefficients and total-item correlations for both the overall scale and its subscales. This result suggests that the items within each subscale measure a cohesive construct. While the average inter-item correlation of 0.42 may indicate a weaker relationship among items, this correlation can be attributed to the multidimensionality of the PBQ, which includes distinct factors like grief, guilt, and anger. Lower correlations among items across different subscales are expected. The split-half reliability further supports the scale’s consistency, indicating that the PBQ produces consistent results across different item groupings.

Our findings also confirmed the concurrent validity of the PBQ-C. The significant positive correlations with the ICG and the depression subscale of the DASS-21 demonstrate that the PBQ captures relevant aspects of grief and depression in the context of pet loss. This finding aligns with the original PBQ study, which found that depressive symptoms may be present in individuals experiencing pet bereavement [22]. Moreover, the significant association with the ICG supports previous research suggesting that levels of grief following a pet’s death can be comparable to those following the loss of a human [38].

The factor analysis of the PBQ-C revealed three distinct factors across 16 items, closely aligning with the original PBQ structure. This consistency suggests that the core dimensions of pet bereavement seem to be applicable across different cultural contexts. However, items 4, 14, and 15 were rearranged into different factors compared to the original structure. Items 4, 14, and 15 were reassigned based on cultural factors specific to the Chinese context.

In particular, item 4, originally categorized under the trauma dimension, was combined with the anger dimension in the original study after the final factor analysis. In this study, it was moved to the guilt dimension. This reassignment is justified by the nature of the item, as cognitive indicators like negative dreams are commonly associated with guilt during the grieving process [39]. Item 14, which involves haunting memories of the pet’s last moments, was reclassified from anger to grief in this study. While such memories could evoke anger, they are also strongly associated with sorrow, loss, and emotional pain, which are central aspects of grief [40]. In Chinese culture, the concept of ‘Lian (連)’, which refers to a deep sense of attachment and emotional continuity even after loss, reinforces this alignment of memories with grief over anger [41]. The cultural emphasis on preserving emotional connections and respecting the significance of past relationships further supports the reassignment of this item. Future studies should explore how these cultural elements influence the expression and experience of grief in pet loss. Categorizing item 15 into the “anger” dimension was consistent with cultural norms in Chinese Society, where expressing anger openly is often discouraged due to social expectations of emotional restraint [42]. The internalization of anger, rather than its direct expression, could reflect unresolved grief, explaining the item’s significant loading onto the anger dimension. This cultural tendency suggests that further research is needed to understand how the suppression of anger influences pet bereavement experiences in Chinese society.

The inclusion of a “not applicable” option for item 1 was a key adaptation in this study. This modification has proven its relevance, with 16% of participants selecting this option. This option addresses the fact that direct interactions with veterinarians may not be common for certain pet owners, ensuring the questionnaire’s applicability across diverse pet ownership experiences. Future iterations of the PBQ in similar contexts should consider maintaining this option to ensure the tool’s accuracy in capturing the full range of bereavement experiences.

The demographic analysis revealed several associations between the examined factors with the PBQ scores. First, the results showed a weak negative correlation between age and guilt, indicating that older individuals reported less guilt. This is consistent with the original study, which also reported a weak negative correlation with guilt [22]. As discussed by Erikson [43], older adults often view death as a natural part of life, leading to a more neutral acceptance and reduced feelings of guilt. A study in the mainland China supports this, showing that older people tend to perceive death as a part of life, which may reduce their sense of responsibility for their pet’s death [44].

No significant correlation was found between age and depressive symptoms, consistent with the original study’s findings. A recent meta-analysis reported that depression among older adults is primarily influenced by factors such as inadequate pension schemes, limited access to healthcare, and declining physical health [45]. These socio-economic challenges may drive depressive symptoms in older adults, even as their grief symptoms decrease over time, suggesting that depression may be influenced more by external factors than by grief alone.

Consistent with the Portuguese PBQ study, our results demonstrated that participants with a greater number of pets had higher levels of anger and grief [25]. The association may be due to the unrecognized grief by the society [18,21]. This lack of recognition makes it common for those around the grieving owners to encourage them to get a new animal soon after the death of their beloved pet [18]. This theme of misrecognition also resonated in a Hong Kong-based study, where the grief process was found to be prolonged due to unfulfilled societal expectations [21]. For owners with multiple pets, the loss of one pet may not be seen as significant, causing their grief to be undervalued by others and resulting in insufficient emotional support and validation. This lack of support and validation may lead to an increased intensity of grief symptoms.

In terms of types of pets, our findings indicate that owners of dogs or cats tend to report greater PBQ scores and experience more profound grief than those with other types of pets, suggesting that losing a dog or cat may have a more substantial emotional effect than losing other kinds of pets. It aligned with previous findings that cat and dog owners were disproportionately in the high grief group compared to other types of pets [46]. These findings are likely due to the a high frequency of interaction and stronger mutual emotional connections typically observed with dogs and cats [47,48].

The analysis also revealed significant differences in PBQ-C scores based on the cause of death, with the highest distress levels associated with poisoning or attacks. This finding aligns with research on ambiguous or unexpected loss, which often intensifies the grieving process due to a lack of closure [49]. In contrast, deaths due to old age generally involve more preparation and acceptance, resulting in lower distress levels. In addition, the anger scores were notably higher in instances of medical negligence or complications after treatment. This feeling could stem from directed anger towards medical professionals or feelings of personal responsibility, as pet owners may struggle with self-blame or question their decisions leading up to the medical outcome [50]. Future studies should investigate these distinct emotional responses to better understand how different causes of death influence pet bereavement.

It is interesting and noteworthy that the participants who had not experienced the euthanasia of their pets reported higher levels of depression, consistent with previous findings suggesting that euthanasia may provide a sense of closure and control, potentially leading to a more resolved grieving process [18,51]. The act of choosing euthanasia may be perceived as a final act of care with a potential of being seen as a more proper way of saying goodbye, and hence may reduce depression levels despite the loss. However, our study did not find significant differences in PBQ scores based on euthanasia experience, contrasting with findings from the Turkish PBQ study, which reported higher guilt among those who avoided euthanasia [24]. This feeling may be due to the difference context in which euthanasia occurs. Sudden or unexpected euthanasia can lead to heightened feelings of anger and guilt; whereas anticipated euthanasia, such as in cases of terminal illness, may allow for mental preparation and acceptance [52,53]. Variations in veterinary support may also play a role. Evidence suggested that the emotional support and communication provided by veterinarians can moderate grief levels during the euthanasia process, especially when reassurance is provided to help ease feelings of guilt [54]. Further studies in Chinese and broader Asian contexts are needed to explore how cultural differences and varying levels of veterinary support impact the grieving process across different regions.

Finally, while the original study found significant associations between time since pet death and marital status with bereavement, our study did not replicate these findings, consistent with results from the Turkish and Portuguese PBQ studies [24,25]. Variability in grief responses may be due to differences in attachment styles, personality traits, and coping mechanisms [19,55], highlighting the complex and individualized nature of the grieving process.

### 4.2. Limitations

This study has several limitations. First, the gender data are absent in the study. The primary focus of this study was to validate and adapt the PBQ for a Chinese context, and as such, gender-specific analysis was not conducted. Although this study did not collect gender-specific data, which limits the scope of our analysis, it is important to acknowledge that previous research has documented significant gender differences in both pet attachment and grief responses. Studies have shown that women tend to form stronger emotional bonds with their pets and experience more intense grief when a pet dies compared to men. This is evident in higher scores on attachment scales such as the Lexington Attachment to Pets Scale (LAPS) and other validated questionnaires [56,57]. Due to their higher attachment level, women often experience more intense grief following the loss of a pet [58].

Gender differences in grief responses have also been well-documented in Chinese society [59,60], suggesting that men and women may experience and express pet bereavement differently. The omission of gender data limits the scope for investigating potential differences in pet bereavement experiences between males and females. Future research should include gender as a demographic variable to provide a more comprehensive understanding of pet bereavement across different genders.

Another limitation is the potential lack of cultural specificity. While the study adapts the PBQ for a Chinese population, it may not fully encompass the diverse cultural attitudes and practices related to pet bereavement across major Chinese regions such as Chian, Singapore, and Taiwan. This limitation is partly due to resource constraints, which restricted the scope of cultural exploration and inclusivity. Having said that, the findings of our previous qualitative study [21] share many similarities of companion animal loss experiences in other countries e.g., US [15], and French-Canada and Japan [61]. To advance the field of understanding companion animal bereavement, future studies should consider conducting very in-depth longitudinal qualitative research to capture a broader range of cultural nuances and incorporate these insights into the questionnaire to enhance its cultural relevance and sensitivity. Third, the study uses convenience sampling with mainly online recruitment through animal welfare and grief platforms. It potentially reaches people with stronger bonding with pets who could have had more intense grief experiences. The future work could benefit from collecting truly representative sampling, such as veterinary practice-based populations and general pet-owner communities. Last but not least, we did not conduct any analyses to examine the discriminant validity and the test-retest reliability of the PBQ-C; it is suggested that if the PBQ is to be validated in other countries in the future, some research teams may include scales that might be able to investigate the discriminant validity and to find a small subset of participants who may be interested in completing in the PBQ about four weeks after the death of a pet so that the test-retest reliability can be examined.

### 4.3. Implications

Despite the limitations, this study has made substantial progress in adapting and validating the Chinese version of PBQ that can be used by individuals who can read Traditional Chinese for assessing the grief associated with pet loss. Also, the PBQ-C provides a culturally sensitive measure for studying pet loss in Chinese populations who are familiar with traditional Chinese that is contributing to some cross-cultural comparisons of grief processes and extending existing literature on pet bereavement. Before this study was conducted, only one qualitative study had explored pet loss and post-bereavement growth within a Chinese context [21], and, at present, with a validated PBQ-C, it is assumed that more pet-loss-related basic quantitative and evaluation studies can be conducted to further our understanding of the pet loss experiences in a much larger Chinese population.

Currently, pet bereavement support services in Chinese societies remain very limited. Existing services primarily include emotional support hotlines and peer support groups, but standardized interventions or those supported by empirical evidence are lacking. The PBQ-C provides a valuable tool that can be used to develop and evaluate targeted interventions, paving the way for evidence-based approaches to support individuals experiencing pet loss.

To better equip or even provide early support to the bereaved pet owners, we suggest that the PBQ-C can be integrated into daily veterinary practices as a screening measure to identify individuals at risk of immediate and prolonged severe grief after the loss of a pet. By incorporating the PBQ-C into routine veterinary care, veterinary professionals can better assess the emotional state of pet owners and offer referrals for further psychological support to promote a more holistic care to both the animals and their owners. Additionally, the development of guidelines and recommendations can aid veterinary professionals in recognizing signs of intense grief and referring the bereaved owners in needs for appropriate emotional support. This approach fosters collaboration between the veterinary and psychological communities, creating a bridge that ensures pet owners receive comprehensive care, from the emotional support provided during veterinary consultations to specialized counseling services when necessary. These skills may add to the sense achievement for veterinary professionals to combat their often stressful and psychologically challenging career [9].

Last but not least, the PBQ-C can be used as a self-screening tool and serve as a resource for public education on grief related to companion animal loss. Given the prevalence of disenfranchised grief in Chinese societies, incorporating pet loss into life and death education can help normalize the experience and provide preventive measures. By utilizing the PBQ-C as part of these educational efforts, individuals can better understand their emotional responses and assess whether they might benefit from professional support. This tool can empower individuals to recognize the signs of unresolved grief and take proactive steps to seek help when needed, reducing the stigma and isolation often associated with grieving a pet.

## 5. Conclusions

In conclusion, the PBQ-C not only offers a culturally tailored measure for understanding pet bereavement but also lays a foundation for advancing research and clinical practices for pet loss in Chinese societies. Through its integration into both support services and public education, we hope to foster a more empathetic and well-informed approach to addressing grief in pet loss especially in the context that many families are not having children and regarding companion animals as their family members.

## Figures and Tables

**Figure 1 animals-14-02845-f001:**
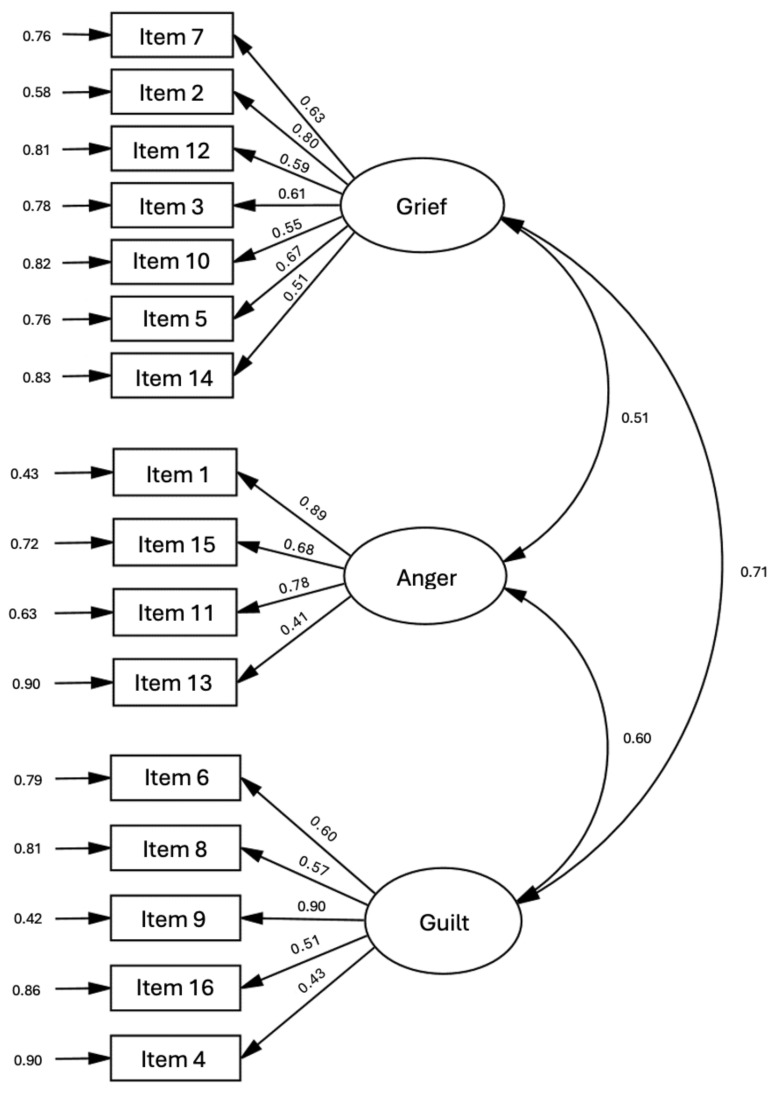
Factor loadings of CFA in PBQ-C (Standardized parameters estimates) with residual errors.

**Table 1 animals-14-02845-t001:** Demographics variables.

	Total (n = 246)	%
Education Level		
No Formal Education	2	0.8
Primary school or below	2	0.8
Junior secondary (Form 1 to Form 3)	17	6.9
Senior secondary (Form 4 to Form 7)/Hong Kong Diploma of Secondary Education	36	14.6
Non-degree courses (including Certificate/Diploma/Higher Diploma/Associate Degree/Yi Jin Diploma)	28	11.4
Bachelor’s degree	88	35.8
Master’s degree or above	73	29.7
Occupation		
Student	21	8.5
Full-time	170	69.1
Part-time	23	9.3
Retired	11	4.5
Housewife	11	4.5
Unemployed/Job-seeking	9	3.7
Other	1	00.4
Marital Status		
Single, without a partner	58	23.6
Single, with a partner	61	24.8
Cohabiting	25	10.2
Married	92	37.4
Divorced/Separated	8	3.3
Widowed	2	0.8
Do you currently have any pets?		
Yes	145	58.9
No	101	41.1
Type of pet lost		
Cat	75	30.5
Dog	110	44.7
Hamster	18	7.3
Rabbit	12	4.9
Bird	8	3.3
Lizard	4	1.6
Other (e.g., Turtle, snail, snake, fish, Guinea Pig, frog, Totoro)	14	5.7
Missing	5	2.0
Cause of pet’s death		
Medical negligence or complications after treatment	28	11.4
Health issues related to old age	163	66.3
Poisoning/attack	10	4.1
Accident	13	5.3
Unknown	24	9.8
Missing values	8	3.3
Have you ever made the decision to have your pet euthanized?		
Yes	95	38.6
No	143	58.1
Missing	8	3.3

**Table 2 animals-14-02845-t002:** Results of Exploratory Factor Analysis (N = 246) with Means, SD, Skewness, and Kurtosis.

Item		Factors Communities			Means (SD)	Skewness	Kurtosis
		Grief	Anger	Guilt			
7	I miss my pet enormously. 我非常想念我的寵物。	0.81			2.43 (0.71)	−1.20	1.18
2	I am very upset about my pet’s death. 我對寵物的離世感到非常痛心。	0.79			2.55 (0.67)	−1.35	1.21
12	I am very sad about the death of my pet. 我對寵物的離世感到非常悲傷。	0.75			2.19 (0.89)	−0.84	−0.20
3	My life feels empty without my pet. 失去寵物後, 我感到生活變得空虛。	0.70			1.96 (0.78)	−0.40	−0.26
10	I cry when I think about my pet. 當我想起我的寵物時, 我會哭泣。	0.61			1.00 (1.02)	0.72	−0.63
5	I feel lonely without my pet. 失去寵物後, 我感到非常孤獨。	0.60			1.77 (0.87)	−0.06	−0.87
14	Memories of my pet’s last moments haunt me. 我對憶起寵物臨終前的最後片段感到非常煎熬。	0.54			1.91 (0.96)	−0.48	−0.76
1	I feel angry at the veterinarian for not being able to save my pet. 我對獸醫無法救活我的寵物而感到憤怒。		0.84		1.28 (1.02)	0.47	−0.85
15	I’ll never get over the loss of my pet. 我永遠無法走出失去寵物的陰影。		0.70		1.41 (1.04)	0.20	−1.15
11	I am angry at other people for contributing to the death of my pet. 我對其他人有份造成我寵物的死亡而感到憤怒。		0.63		1.54 (0.98)	0.15	−1.04
13	I am angry at my friends/family for not being more helpful. 我對我的朋友/家人沒有提供更多的幫助而感到憤怒。		0.56		0.80 (0.85)	0.95	0.37
6	I should have known that something bad could have happened to my pet. 我本應可以察覺我的寵物可能會發生不好的事情。			0.73	1.83 (0.90)	−0.42	−0.53
8	I feel very guilty for not taking care better care of my pet. 我對於未能給予我的寵物更好的照顧而感到非常內疚。			0.68	1.99 (0.99)	−0.52	−0.66
9	I feel bad that I didn’t do more to save my pet. 我對於我沒有做更多的事情來拯救我的寵物而感到遺憾。			0.63	1.85 (0.99)	−0.27	−1.10
16	I wish I had shown my pet more love. 我希望我能對我的寵物表現出更多的愛。			0.60	2.40 (0.74)	−1.16	1.09
4	I have had nightmares about my pet’s death. 我曾經做過關於我寵物離世的噩夢。			0.43	1.38 (0.92)	0.13	−0.79
Cumulative Explained Variance Ratio		23%	46%	63%			

**Table 3 animals-14-02845-t003:** Fit indices of the models.

	χ^2^/df	CFI	NFI	GFI	RMSEA	AIC
Model 1	3.93	0.86	0.82	0.81	0.09	389.09
Model 2	2.26	0.94	0.90	0.92	0.05	298.50

χ^2^/df: Chi-Square divided by Degrees of Freedom, CFI: Comparative Fit Index, NFI: Norm Fit Index, GFI: Goodness of Fit Index, RMSEA: Root Mean Square Error of Approximation, AIC: Akaike Information Criterion.

**Table 4 animals-14-02845-t004:** Correlation between PBQ and DASS Depression and ICG (N= 246).

Measures	Total PBQ	Grief	Guilt	Anger
ICG	0.78 **	0.70 **	0.62 **	0.73 **
DASS Depression	0.71 **	0.62 **	0.59 **	0.68 **
Total PBQ	-	0.92 **	0.88 **	0.81 **
Grief	0.92 **	-	0.71 **	0.62 **
Guilt	0.88 **	0.71 **	-	0.61 **
Anger	0.81 **	0.62 **	0.61 **	-

** *p* < 0.001.

**Table 5 animals-14-02845-t005:** Reliability Analysis in item–total correlation and Cronbach Alpha Coefficient if item deleted (N = 246).

Items		Item-Total Correlation	Cronbach’s Alpha Value When the Item Is Deleted
1	I feel angry at the veterinarian for not being able to save my pet. 我對獸醫無法救活我的寵物而感到憤怒。	0.53	0.92
2	I am very upset about my pet’s death. 我對寵物的離世感到非常痛心。	0.52	0.92
3	My life feels empty without my pet. 失去寵物後, 我感到生活變得空虛。	0.69	0.92
4	I have had nightmares about my pet’s death. 我曾經做過關於我寵物離世的噩夢。	0.46	0.92
5	I feel lonely without my pet. 失去寵物後, 我感到非常孤獨。	0.75	0.91
6	I should have known that something bad could have happened to my pet. 我本應可以察覺我的寵物可能會發生不好的事情。	0.48	0.92
7	I miss my pet enormously. 我非常想念我的寵物。	0.57	0.92
8	I feel very guilty for not taking care better care of my pet. 我對於未能給予我的寵物更好的照顧而感到非常內疚。	0.71	0.91
9	I feel bad that I didn’t do more to save my pet. 我對於我沒有做更多的事情來拯救我的寵物而感到遺憾。	0.73	0.91
10	I cry when I think about my pet. 當我想起我的寵物時, 我會哭泣。	0.70	0.91
11	I am angry at other people for contributing to the death of my pet. 我對其他人有份造成我寵物的死亡而感到憤怒。	0.53	0.92
12	I am very sad about the death of my pet. 我對寵物的離世感到非常悲傷。	0.70	0.91
13	I am angry at my friends/family for not being more helpful. 我對我的朋友/家人沒有提供更多的幫助而感到憤怒。	0.48	0.92
14	Memories of my pet’s last moments haunt me. 我對憶起寵物臨終前的最後片段感到非常煎熬。	0.74	0.91
15	I’ll never get over the loss of my pet. 我永遠無法走出失去寵物的陰影。	0.74	0.91
16	I wish I had shown my pet more love. 我希望我能對我的寵物表現出更多的愛。	0.56	0.92

## Data Availability

The data supporting this study’s findings are available from the corresponding author upon reasonable request.

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
