# Peer review of "Adaptation and Validation of the Pet Bereavement Questionnaire (PBQ) for Chinese Population"

_animals, 2024, doi:10.3390/ani14192845_

Round 1

Reviewer 1 Report

Comments and Suggestions for Authors

This paper presents a study of the adaptation of an instrument for assessing bereavement due to the death of pets for the Chinese population, which is an innovative contribution in that specific cultural context and in a field that is proving to be relevant. Overall, it is a clearly written article in which the procedure adopted to conduct the study, as well as the data analysis, are presented rigorously.

Minor revisions are suggested:

1- When describing Participants, the authors classify them in terms of “race”. The information on this aspect should be removed. Not only is there little consensus on the concept of race, but concepts of race and nationality seem to be confused in the data presented. As this information is not relevant to the psychometric study carried out (what is important is Chinese cultural context), it is suggested that it be removed from the article.

2- Authors carry out a very careful psychometric study, presenting the results of the analysis of factorial validity, internal and external construct validity, reliability and sensitivity. With regard to these analyses, it is only suggested that, when they present the item-total correlation data in the text (lines 265 to 267) and in the table (Table 5), they make it clear if that these correlations have been corrected for overlap (if they are not, it is suggested to correct them).

3- This psychometric analysis could be enriched by a description of the responses to each of the items - minimum and maximum values, mean, median, asymmetry and kurtosis. This analysis would allow a more in-depth understanding of the instrument's sensitivity.

4- In the sentence presented in lines 277 to 278 (“It was not...”) it is not clear what the subject of the sentence is. Are you referring to age?

5- Lines 280 to 284 show the median for the total PBQ, but not for the Grief subscale. It is suggested adding the median for the Grief subscale.

Author Response

Comment 1- When describing Participants, the authors classify them in terms of “race”. The information on this aspect should be removed. Not only is there little consensus on the concept of race, but concepts of race and nationality seem to be confused in the data presented. As this information is not relevant to the psychometric study carried out (what is important is Chinese cultural context), it is suggested that it be removed from the article.

Response:  Thank you for the comment. We agree and deleted the “race-related” information

Comment 2- Authors carry out a very careful psychometric study, presenting the results of the analysis of factorial validity, internal and external construct validity, reliability and sensitivity. With regard to these analyses, it is only suggested that, when they present the item-total correlation data in the text (lines 265 to 267) and in the table (Table 5), they make it clear if that these correlations have been corrected for overlap (if they are not, it is suggested to correct them).

Response: Thank you for this comment. The item was calculated without overlapping (e.g., item 1 is correlated with item 2 – 10), added “corrected for overlap” (line 375)

Comment 3- This psychometric analysis could be enriched by a description of the responses to each of the items - minimum and maximum values, mean, median, asymmetry and kurtosis. This analysis would allow a more in-depth understanding of the instrument's sensitivity.

Response: The requested information was added the information in Table 2.

Comment 4- In the sentence presented in lines 277 to 278 (“It was not...”) it is not clear what the subject of the sentence is. Are you referring to age?

Response: Thank you for pointing this out and we have changed “it” to “age” (line 391)  

Comment 5- Lines 280 to 284 show the median for the total PBQ, but not for the Grief subscale. It is suggested adding the median for the Grief subscale.

Response: Thank you and we added the median in line 396-398

Reviewer 2 Report

Comments and Suggestions for Authors

The manuscript offers a Chinese translation of the pet bereavement questionnaire (PBQ) and examines psychometric properties with a Hong Kong sample (N=245). The steps taken for validation are generally relevant, and divergences from the original in terms of factor structure are convincingly discussed as possible effect of cultural difference. I congratulate the authors for contributing this pet bereavement assessment tool, but some changes to the manuscript are warranted, see below.

The introduction (38-66) sets the stage by describing positive effects of having a companion animal and mentions association with attachment and risk for long term grief – all of which is relevant, but the authors’ overstate their case – please provide a more nuanced picture:

- Some sources (7,8) are not scientific

- Critique of the so-called pet effect for being exaggerated (e.g., Hal Herzog) is ignored

- Lines 63-66 are unclear. The text easily reads as suggesting that a study found pet grievers to often grieve for more than a year and suffer functional reductions, however, those were not findings of the study (reference 17) but its eligibility criteria for being included in its sample.

BTW, some typos/missing words:

line 60-61: Those who integrate… should probably be: For those who integrate

line 63: A recent has… should probably be: A recent study has

line 75: author’s should be authors’

Title: Should ...for Chinese Population by ...for Chinese populations (plural) or ...for the Chinese Population?

2 Materials and methods

When participants had had more than one pet, how did they choose which one was referred to when assessing their grief? This info is missing. Since species, how it died, and how long ago can all influence the report, it would be nice to know. If no instruction was given, please add to limitations.

Gender info is missing. The authors’ mention this in limitations, which is good, and I don’t think it disqualifies the study for publication. However, it is a problem since levels of both pet attachment and grief vary with gender in some countries, and it should be covered more in the discussion section.

N=245 is not a large sample for CSA but it’s ok here I think.

Lines 198-: Why were nonparametric analyses chosen? Please explain.

While the psychometric analyses that are reported are relevant, some are left out without mentioning so. For example, concurrent validity was assessed by ICG and DASS-Dep, which is relevant, but no discriminant validity measures were employed (and the factor structure slightly changed) so I am unsure about the extent to which findings might reflect general response tendency plus randomness rather than pet grief per se.

Similarly, split-half reliability is good but test-retest reliability could also be relevant.

I don’t see these omissions as disqualifying but addressing them would add quality.

3. Results

Table 1 is extremely long and detailed – I believe some categories could be collapsed. (Since no data on the general population in Hong Kong is provided for comparison, it tells us little anyway).

Also, weren't some of the original data in a form that allows averages (e.g., age, income, kids…)?

If the question was “Have you ever euthanized your pet?” (see end of table1), we don’t know if that was the same pet as they are assessing their grief for (cf my critique of method above), leaving discussions of  euthanasia as possible factor unfounded. Please clarify.

Finally, we need to know the distribution also – min, max, mean, SD of scale and subscales. Please report those.

I understand that the sample is likely to be biased by the self-selection process (as aptly discussed by the authors) and requiring caution. Nevertheless, we need to have some idea of the distribution. This is required (a) to compare to other countries in research, and (b) to use as a crude baseline if the tool is to have clinical/predictive value for clients and veterinarians. Both uses are aims stated by the authors at the outset and in Implications (lines 455-472)

Comments on the Quality of English Language

some typos/omissions are mentioned in the former box. The English is fine. 

Author Response

Introduction

Comment 1 - The introduction (38-66) sets the stage by describing positive effects of having a companion animal and mentions association with attachment and risk for long term grief – all of which is relevant, but the authors overstate their case – please provide a more nuanced picture

Response 1 – Thank you for the comment, we have heavily revised the mentioned paragraphs to minimise the sense of overstating the impacts of pet ownership on human animals.

Comment 2 - Some sources (7,8) are not scientific

Response 2 - Thank you for your insightful comment. We acknowledge that sources (7,8) are subjective, but we wish to cite them because there are limited very recent scientific studies in this topic to state the trend.

Comment 3 - Critique of the so-called pet effect for being exaggerated (e.g., Hal Herzog) is ignored

Response 3 – Please see our response to point one. Thank you.

Comment 4 - Lines 63-66 are unclear. The text easily reads as suggesting that a study found pet grievers to often grieve for more than a year and suffer functional reductions, however, those were not findings of the study (reference 17) but its eligibility criteria for being included in its sample.

Response 4 – Thank you for your valuable feedback on lines 63-66. Upon reviewing the study and your comment, we have revised the sentence to distinguish between the criteria used for participant selection (grieving for a year or more with functional impairment) and the study’s findings, which suggest that the participants’ grieving processes aligned with the DSM-5 model of grief (line 85 – 89).

Typo/missing words

Comment 5 - line 60-61: Those who integrate… should probably be: For those who integrate

Response 5 – Changed (line 83)

Comment 6 – line 63: A recent has… should probably be: A recent study has..

Response 6 – Changed (line 85)

Comment 7 - line 75: author’s should be authors’

Response 7 – Changed (line 98)

Comment 8 - Title: Should ...for Chinese Population by ...for Chinese populations (plural) or ...for the Chinese Population?

Response 8 – This scale applies broadly to the entire Chinese population, we think keeping “Chinese Population” is more appropriate

 Materials and methods

Comment 9 - When participants had had more than one pet, how did they choose which one was referred to when assessing their grief? This info is missing. Since species, how it died, and how long ago can all influence the report, it would be nice to know. If no instruction was given, please add to limitations.

Response 9 – The participants were instructed to select their most recently deceased pet, and we have added this clarification in lines 186-187. Thank you for your valuable feedback.

Comment 10 - Gender info is missing. The authors’ mention this in limitations, which is good, and I don’t think it disqualifies the study for publication. However, it is a problem since levels of both pet attachment and grief vary with gender in some countries, and it should be covered more in the discussion section.

Response 10 – Added details in discussion (line 546-553)

Comment 11 - N=246 is not a large sample for CSA but it’s ok here I think.

Response 11 – Thank you!

Comment 12 - Lines 198: Why were nonparametric analyses chosen? Please explain.

Response 12 - Explained in line 295-297

Comment 13 - While the psychometric analyses that are reported are relevant, some are left out without mentioning so. For example, concurrent validity was assessed by ICG and DASS-Dep, which is relevant, but no discriminant validity measures were employed (and the factor structure slightly changed) so I am unsure about the extent to which findings might reflect general response tendency plus randomness rather than pet grief per se.

Response 13 – Thank you for pointing this out, and we have added in the limitation and state that may be in the future, discriminant validity and test-retest reliability are needed to further enhance the psychometric properties of the PBQ (line 575-581).

Comment 14 - Similarly, split-half reliability is good, but test-retest reliability could also be relevant. I don’t see these omissions as disqualifying but addressing them would add quality.

Response 14 – Thank you and please see above response.

Results

Comment 15 - Table 1 is extremely long and detailed – I believe some categories could be collapsed. (Since no data on the general population in Hong Kong is provided for comparison, it tells us little anyway).

Response 15 – Thank you and we have removed some of the variables and described them only in the text (line 308-314).

Comment 16 - Also, weren't some of the original data in a form that allows averages (e.g., age, income, kids…)?

Response 16 – Thankyou and please see previous response.

Comment 17 - If the question was “Have you ever euthanized your pet?” (see end of table1), we don’t know if that was the same pet as they are assessing their grief for (my critique of method above), leaving discussions of euthanasia as possible factor unfounded. Please clarify.

Response 17 - The participants were instructed to select their most recently deceased pet, and we have added this clarification in lines 186-187. Thank you for your valuable feedback.

Comment 18 - Finally, we need to know the distribution also – min, max, mean, SD of scale and subscales. Please report those.

Response 18 – Thank you and the information were added in Table 2.

Comment 19 - I understand that the sample is likely to be biased by the self-selection process (as aptly discussed by the authors) and requiring caution. Nevertheless, we need to have some idea of the distribution. This is required (a) to compare to other countries in research, and (b) to use as a crude baseline if the tool is to have clinical/predictive value for clients and veterinarians. Both uses are aims stated by the authors at the outset and in Implications (lines 455-472)

Response 19 – Thank you so much for this very insightful comment. Indeed, it would be wonderful if we could have an understanding of the baseline level of what level is general and what needs to be cautious for owners and professionals, unfortunately, as we discussed, there have not been many validation studies of the PBQ and we could only make general comparisons and hoping that more similar studies could be conducted in the nearest future to enhance the implications the PBQ in other cultures. 

Reviewer 3 Report

Comments and Suggestions for Authors

The authors modified and validated the Pet Bereavement Questionnaire for a Chinese population (Hong Kong). Most research on the effects of pet loss on owners concerns Western populations, making the research topic of this paper somewhat unique. The English is perfectly understandable, but there are some places where words are missing or where language can be improved (I point some of these out below). The major limitation was lack of information on the gender of respondents, but the authors address this issue in the Discussion. I especially liked the cultural comparisons of differences between the original PBQ and the PBQ-C in the Discussion. It would be very helpful to readers if the authors included the full PBQ-C (in English) as an Appendix. I am unfamiliar with research guidelines in China, but studies involving questionnaires in the U.S. require approval by a University’s Institutional Review Board – is there something similar in China? If so, you should state that in line 495. Finally, although informed consent was obtained in your study (line 98), you may want to state that again in line 496. My specific comments are detailed below.

Simple Summary:

Given that the Simple Summary is geared toward the lay public, perhaps briefly define the terms “reliability” and “validity” in lines 15-16.

Abstract:

In the same way, some readers interested in pet-human interactions may not be familiar with the terms “internal consistency,” “split-half reliability,” and “concurrent validity.” Perhaps briefly define these as well?

Keywords:

The terms “Pet bereavement” and “Chinese” and “Validation” occur in the title, so they will be picked up during searches. Consider replacing these terms with others such as “depression,” “anxiety,” and “grief.”

Introduction:

Lines 86-88: Thank you for explicitly stating your hypotheses; this might be a good place to briefly define “internal consistency” and “concurrent validity.”

Materials and Methods:

Line 102: Note that there is a minor difference in sample size between this section (n = 245 participants) and the Results section (n = 246 participants; line 210 and Table 1).

Discussion:

Lines 418-426: Did the original PBQ include gender? What about the Portuguese version? It would have been interesting to see if the participants in your study were female-biased, as this is typical of studies on human-non-human animal interactions in Western societies (see Herzog, H. Women dominate research on Human-Animal Bond. Am. Psychol., 2021, May 24. https://www.psychologytoday.com/us/blog/animals-and-us/202105/women-dominate-research-the-human-animal-bond.

Line 432: Which paper does “our previous qualitative study” refer to? Please include the reference number here.

Minor points:

Lines 43-44: Change to “The patterns of companion animal ownership are likely to have recently changed given broad social shifts….”

Line 63: Word missing: A recent study?

Line 108: Change “child” to “children”

Line 275: Delete “was found”

Line 285: Change “were” to “was”

Line 371: Change to “found to be prolonged”

Line 429: Delete “dominate” or change to “major”

Comments on the Quality of English Language

The English is perfectly understandable, but there are some places where words are missing or where language can be improved (I pointed some of these out in my comments to authors).

Author Response

Comment 1: Simple Summary: Given that the Simple Summary is geared toward the lay public, perhaps briefly define the terms “reliability” and “validity” in lines 15-16.
Response 1: Thank you, and we added a simple definition (line 11-13).

Comment 2: Abstract: In the same way, some readers interested in pet-human interactions may not be familiar with the terms “internal consistency,” “split-half reliability,” and “concurrent validity.” Perhaps briefly define these as well?
Response 2: Thank you, and we added a brief definition in the abstract (line 27-31).

Comment 3: Keywords: The terms “Pet bereavement” and “Chinese” and “Validation” occur in the title, so they will be picked up during searches. Consider replacing these terms with others such as “depression,” “anxiety,” and “grief.”
Response 3: Deleted “pet” and “Chinese validation,” added “grief” and “depression.” Thank you.

Comment 4: Introduction: Lines 86-88: Thank you for explicitly stating your hypotheses; this might be a good place to briefly define “internal consistency” and “concurrent validity.”
Response 4: Added definition, thank you. (line 164-167).

Comment 5: Materials and Methods: Line 102: Note that there is a minor difference in sample size between this section (n = 245 participants) and the Results section (n = 246 participants; line 210 and Table 1).
Response 5: Thank you for spotting this, and we have recalculated.

Comment 6: Discussion: Lines 418-426: Did the original PBQ include gender? What about the Portuguese version? It would have been interesting to see if the participants in your study were female-biased, as this is typical of studies on human-non-human animal interactions in Western societies (see Herzog, H. Women dominate research on Human-Animal Bond. Am. Psychol., 2021, May 24. https://www.psychologytoday.com/us/blog/animals-and-us/202105/women-dominate-research-the-human-animal-bond).
Response 6: The original PBQ, as well as other versions like the Portuguese version, did include gender data; however, it was an oversight in our study not to collect this information. While gender-specific analysis was not the focus of this study, we acknowledge that including gender would have provided additional insights, particularly given the documented gender differences in grief responses. We will ensure that future studies incorporate gender data to better explore potential differences in pet bereavement experiences. Thank you for spotting this, and we fully acknowledge that gender has been suggested to have some difference in the bereavement experiences.

Comment 7: Line 432: Which paper does “our previous qualitative study” refer to? Please include the reference number here.
Response 7: Added reference number, thank you.

Minor points:
Comment 8: Lines 43-44: Change to “The patterns of companion animal ownership are likely to have recently changed given broad social shifts….”
Response 8: Changed, please refer to line 63-64, thank you.

Comment 9: Line 63: Word missing: A recent study?
Response 9: Added, please refer to line 85.

Comment 10: Line 108: Change “child” to “children.”
Response 10: Changed, and please refer to line 190.

Comment 11: Line 275: Delete “was found.”
Response 11: Deleted.

Comment 12: Line 285: Change “were” to “was.”
Response 12: Changed.

Comment 13: Line 371: Change to “found to be prolonged.”
Response 13: Added “be.”

Comment 14: Line 429: Delete “dominate” or change to “major.”
Response 14: Changed.

Reviewer 4 Report

Comments and Suggestions for Authors

The article presents a well conducted study designed to adapt and validate a pet bereavement questionnaire to be culturally appropriate for use among Chinese populations.  

There are a few language issues which will be commented on in the next section of this review. 

Introduction

lines 63-67: It is not clear what the authors are saying in this sentence.

line 76 "explores grief reaction and post-traumatic growth"  What is post-traumatic growth?

Depression subscale (DASS-21)

lines 127-136:  Did the authors compute Cronbach's alpha for their study population?  This would be useful.

Inventory of Complicated Grief

lines 137-146: Did the authors compute Cronbach's alpha for their study population? This would be useful.

Methods/Results

It would be helpful to explain how the participants were asked to report the pet they lost.  It is not clear if there were any directions regarding selecting the pet who died, therefore it is not clear how to interpret the questions regarding length of time since the loss of the pet, the cause of death, and the experience with euthanasia.   How do you know if the responses received were for the same pet?  

line 277: "It was not"  Do you mean age was not?

Comments on the Quality of English Language

line 40: "there has been a rising 72 per cent.."  There has been an increase of 72 per cent

line 70: "Under the Chinese context,"   In the Chinese context,

line 74: "as research in exploring pet bereavement within Chinese societies were very limited"  was very limited

line 108: "number of child"  number of children

line 110-111: "whether they have undergone euthanasia for their deceased pet" whether their deceased pet was euthanized

line 189: "due to small sample size"  due to the small sample size

line 276: "and guilt scale" and the guilt scale 

line 285: "There were a significant difference"  There was a significant different

line 285-287: "who had experienced euthanasia"  whose pets had been euthanized  and "with individuals who had experienced euthanasia reported" with individuals whose pets had been euthanized reporting

line 338 "into "anger" dimension was consistent with cultural norm"  into the "anger" dimension was consistent with cultural norms

line 371: " was found to prolonged"  was found to be prolonged

line 402: "the difference context"  the different context

line 418 "the gender data has absence in the study"  We did not collect gender data in the study.

line 420 "However, Gender"  However, gender

Author Response

Introduction

Comment 1 - lines 63-67: It is not clear what the authors are saying in this sentence.
Response 1 - Rephrased the sentence (line 85-89), thank you.

Comment 2 - Line 76: "explores grief reaction and post-traumatic growth" What is post-traumatic growth?
Response 2 - Added definition in (line 99-100), thank you.

Depression Subscale (DASS-21)

Comment 3 - lines 127-136: Did the authors compute Cronbach's alpha for their study population? This would be useful.
Response 3 - Added Cronbach's alpha for the original scale (line 217-218).

Inventory of Complicated Grief

Comment 4 - lines 137-146: Did the authors compute Cronbach's alpha for their study population? This would be useful.
Response 4 - Added Cronbach's alpha for the original scale (line 277-278).

Methods/Results

Comment 5 - It would be helpful to explain how the participants were asked to report the pet they lost. It is not clear if there were any directions regarding selecting the pet who died; therefore, it is not clear how to interpret the questions regarding the length of time since the loss of the pet, the cause of death, and the experience with euthanasia. How do you know if the responses received were for the same pet?
Response 5 - Added explanation (line 186-187), thank you for pointing this out.

Comment 6 - Line 277: "It was not" Do you mean age was not?
Response 6 - Changed “it” to "the age," thank you for pointing this out.